# Distinguishing mature and immature trees allows to estimate forest carbon uptake from stand structure

Samuel M. Fischer[1,2], Xugao Wang[3], and Andreas Huth[1,2,4]

[1]Helmholtz Centre for Environmental Research – UFZ, Dept. of Ecological Modelling, Permoserstr. 15, 04318 Leipzig, Germany.

[2]Osnabrück University, Institute of Environmental Systems Research, Barbarastr. 12, 49076 Osnabrück, Germany.

[3]Chinese Academy of Sciences, Institute of Applied Ecology, PO Box 417, Shenyang 110016, China.

[4]German Centre for Integrative Biodiversity Research (iDiv) Halle-Jena-Leipzig, Puschstr. 4, 04103 Leipzig, Germany.

**Correspondence:** Samuel M. Fischer (samuel.fischer@ufz.de)

**Abstract.** Relating forest productivity to local variations in forest structure has been a long-standing challenge. Previous studies often focused on the connection between forest structure and stand-level photosynthesis (GPP). However, biomass production (NPP) and net ecosystem exchange (NEE) are also subject to respiration and other carbon losses, which vary with local conditions and life history traits. Here, we use a simulation approach to study how these losses impact forest productivity and reveal themselves in forest structure. We fit the process-based forest model Formind to a $25\,\mathrm{ha}$ inventory of an old-growth temperate forest in China and classify trees as "mature" (full-grown) or "immature" based on their intrinsic carbon use efficiency. Our results reveal a strong negative connection between the stand-level carbon use efficiency and the prevalence of mature trees: GPP increases with the total basal area, whereas NPP and NEE are driven by the basal area of immature trees. Accordingly, the basal area entropy – a structural proxy for the prevalence of immature trees – correlated well with NPP and NEE and had a higher predictive power than other structural characteristics such as Shannon diversity and height standard deviation. Our results were robust across spatial scales ($0.04$-$1\,\mathrm{ha}$) and yield promising hypotheses for field studies and new theoretical work.

**Keywords:** carbon balance, carbon use efficiency, forest structure, modelling, primary production

## 1 Introduction

Understanding the drivers of forest productivity is key for assessing forests' ability to provide ecosystem services (e.g. carbon sequestration or commercial wood production) and to gauge their resilience against disturbances and global change (Costanza

et al., 1998; Anav et al., 2015; Jha et al., 2019; Sheil and Bongers, 2020). Forests' net primary production (NPP) may be affected via two pathways: carbon supply, i.e., gross primary production (GPP), and carbon losses due to respiratory costs and other limiting factors (Wiley and Helliker, 2012). Forest structure (e.g. density, species composition, age and size distribution; McElhinny et al., 2005) can be both a factor and result of processes acting on either of these pathways (Waide et al., 1999; Forrester and Bauhus, 2016; Sheil and Bongers, 2020). For example, denser forests may exhibit a larger total leaf area and hence higher stand productivity. Conversely, high productivity of individual trees may lead to denser forests. Hence, identifying the connection between forest structure and productivity is key for a comprehensive understanding of forest productivity.

Several studies have established links between forest structure and carbon supply (Waide et al., 1999; Forrester and Bauhus, 2016). For example, GPP is expected to benefit from higher diversity via improved exploitation of ecological niches and reduced competition, and vertically stratified forests may allow for more efficient light use due to denser leaf packaging (Forrester and Bauhus, 2016; Bohn and Huth, 2017). Nonetheless, it has proven difficult to identify clear relationships between forest structure and NPP (Chisholm et al., 2013), as diverse factors, ranging from resource availability to the impact of biotic agents, affect forest dynamics on different procedural levels (Forrester and Bauhus, 2016), and NPP is not only subject to supply-related but also loss-related factors. A unified framework for forest productivity therefore also needs to address the corresponding role of losses. This is the subject of this study.

A tree's ability to utilize acquired carbon to form biomass can be expressed through its carbon use efficiency (CUE = NPP/GPP). In the absence of shading by larger plants, the CUE is expected to decline with tree size, as larger trees have a higher demand for respiration and non-structural carbon (Collalti et al., 2020b; Binkley, 2023). Such respiratory losses and other, external, factors may induce site-dependent tree size maxima, at which biomass accumulation is significantly reduced. The resulting decline of NPP with forest age is well documented on the stand level (Gower et al., 1996; Tang et al., 2014; Collalti et al., 2020a), but the extent at which loss-induced limitations drive variations of NPP on the local scale is less understood (Chisholm et al., 2013; Rödig et al., 2018). This, however, would be necessary for a mechanistic understanding of the impact of loss-related factors in comparison to supply-related factors.

To evaluate the impact of loss-induced limitations on forest productivity, we suggest a simple classification framework: we divide trees into full-grown (below: "mature") and growing ("immature") trees based on their intrinsic optimal CUE, i.e., the CUE the trees could attain if their GPP was not limited by competition. We consider trees as mature if intrinsic loss-related factors limit their CUE even under otherwise optimal growth conditions. Consequently, tree maturity and competition are distinct processes reducing stand-level forest productivity.

Forest productivity may be considered on different procedural levels: GPP, representing forests' photosynthetic capacity; NPP, denoting their total wood production after respiratory losses; and the net ecosystem exchange (NEE), measuring the total forest carbon sequestration in the presence of emissions from deadwood decomposition and soil respiration. Studying the impact of loss-induced growth limits, we focused on three questions:

1. How do GPP, NPP, and NEE depend on the prevalence of mature and immature trees?

2. How can these relationships be linked to forest structure and expressed via easily measurable forest characteristics?

3. On which spatial scales can these relationships be observed?

To answer these questions, local carbon fluxes must be identified. Though NPP may be estimated from forest inventory data, field data for GPP and NEE, e.g. from eddy covariance measurements, are typically only available for larger scales (about $10\,\mathrm{ha}$). Similarly, it can be difficult to determine which trees have reached the mature stage. These challenges can be addressed with process-based forest models, which reproduce the forest dynamics under controlled reference conditions and provide full insight into carbon fluxes as well as the state and growth limitations of each tree.

There is a broad variety of forest models covering diverse sets of processes potentially impacting forest dynamics (Bugmann and Seidl, 2022). Depending on their respective main use cases, the models differ in their spatial resolution, their representation of vertical forest structure, physiological detail, and consideration of abiotic (e.g. soil conditions, weather, fire) and biotic (e.g. browsing, bark beetle attacks) factors (Merganičová et al., 2019; Bugmann and Seidl, 2022).In this study, we used the individual-based forest gap model FORMIND (Bohn et al., 2014; Fischer et al., 2016). The model features submodels on regeneration, competition, growth, and mortality and has been applied to study forest dynamics and carbon fluxes in a variety of both temperate and tropical forests (Fischer et al., 2016). As the model represents individual trees and the forest's vertical leaf distribution explicitly, FORMIND is particularly suited for studying the relationship between forest structure and forest productivity (Bohn and Huth, 2017). At the same time, the gap model approach of aggregating the impacts of individual trees at the local level leads to relatively high computational efficiency in large-scale simulations (Shugart et al., 2018).

We parameterized the model to mimic the dynamics of a species-rich old-growth temperate forest in Changbaishan, China. Located in a natural reserve, this forest offers unique opportunities to study long-term forest dynamics without biases introduced by human interventions. We addressed the research questions by computing GPP, NPP, and NEE on different spatial scales ($0.04\,\mathrm{ha}$ and $1\,\mathrm{ha}$) and setting them into relation with the basal area of mature and immature trees as well as different measures for structural diversity. For question (2), we suggest the DBH entropy, a measure for the diversity of tree heights, as a general proxy for the prevalence of immature trees and therefore also forest productivity.

## 2 Materials and Methods

We applied a data-driven modelling approach (Fig. 1) to analyze the relationship between forest structure and forest productivity. We fitted the process-based forest model FORMIND to forest inventory data from Changbaishan, China, and data on species' traits and allometric relationships. Using the model, we then linked forest productivity to the prevalence of mature trees and other forest characteristics. Below we describe the individual steps in detail.

### 2.1 Field data

We based our analysis on forest inventory data from an old-growth temperate forest in the Changbaishan National Nature Reserve in northeastern China. The surveyed area consists of $25\,\mathrm{ha}$ of conifer/broad-leaf mixed forest with $47$ species, a total biomass of $302^{\,\mathrm{t\,ODM}}/_{\mathrm{ha}}$ (Piponiot et al., 2022). The inventory data contain the position, diameter at breast height (DBH) and species of each tree with $\mathrm{DBH} \geq 1\mathrm{cm}$ for the census years 2004, 2009, and 2014. Each tree is uniquely identified with an ID

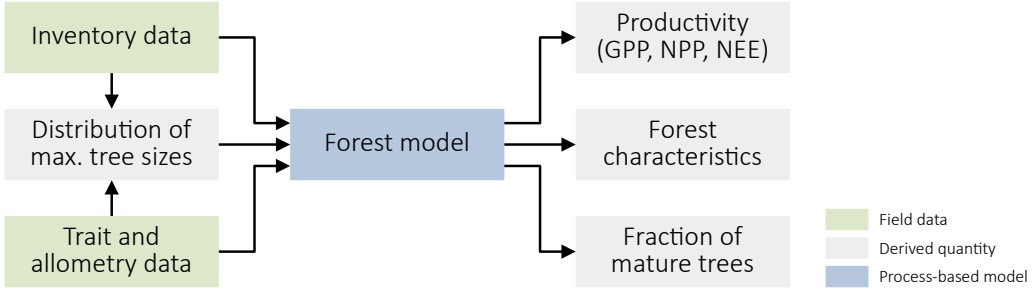

**Figure 1.** Summary of our approach. We use forest inventory data and data on species' traits and allometric relationships to derive the distribution of maximal plant sizes and parameterize a process-based forest model. This model, in turn, yields productivity metrics (GPP, NPP, and NEE) and different forest characteristics, including the fraction of mature trees.

number. For trees that had multiple stems at breast height, we focused on the main stem (maximal DBH) in our analysis and we disregarded minor stems.

In addition to the inventory data, we used information on traits and allometry of the species from field measurements. These data included DBH-dependent heights, crown radii and crown base heights. Furthermore, the dataset included the species' wood densities and shade tolerance types ("light demanding", "mid-tolerant", or "shade tolerant"). Not all of these data were available for all species; we provide details in Supplementary Information (SI) A.

### 2.2   Model and parameterization

FORMIND is a process-based forest gap model featuring the main processes regeneration, competition, tree growth, and mortality (Fischer et al., 2016). Trees are mainly characterized by their DBH and species. Other properties, such as plant height or crown size, are derived from the DBH via allometric relationships. The model considers $20\,\mathrm{m} \times 20\,\mathrm{m}$ forest patches, for which the vertical leaf distribution and the resulting light climate are computed. The obtained incident radiation is used to compute each tree's GPP. The corresponding NPP is computed by subtracting an individual's respiration and other carbon losses from

its GPP. Here, the maintenance respiration is determined by comparing the estimated GPP of trees under unshaded reference conditions with corresponding biomass increments from field data. Growth respiration and other carbon losses are computed as a certain fraction of the difference between GPP and maintenance respiration.

In the past, FORMIND was parameterized for managed European temperate forests (Bohn et al., 2014), but the Changbaishan forest has a different, richer species pool and is old-growth, requiring a correspondingly parameterized regeneration module.

Therefore, we needed to develop an adjusted parameterization to apply FORMIND to this site. Below we summarize how we parameterized the model and highlight changes to the version described before in Fischer et al. (2016). Details can be found in SI B.

## Basic parameterization

To reduce model complexity in the species-rich Changbaishan setting, we aggregated species into plant functional types (PFTs) based on their maximal DBHs (below / above $30\,\mathrm{cm}$) and light demand (light demanding, mid-tolerant, and shade tolerant). When data necessary for the classification were not available, we assigned species via a likelihood-based cluster analysis based on shade tolerance (Niinemets and Valladares, 2006; Wang et al., 2010) and observed tree growth (SI B2). Because *Quercus mongolica* had a significantly different size structure than the other light-demanding species, we divided the large light demanding into two PFTs, one with all other large light demanding species and one for *Q. mongolica* only. We obtained six PFTs: small light demanding, large light demanding 1 and 2, large mid-tolerant, small shade tolerant, and large shade tolerant species. There were no small mid-tolerant species.

We estimated mean traits and allometric relationships for the PFTs based on the trait and allometry data. When computing the means, we weighted species according to their shares in the inventory to best reflect the species composition in the study area. Details can be found in SI B3 and B4. We modelled the forest under constant climatic conditions, which we derived based on data from the literature (evapotranspiration: Sun et al., 2004; temperature: Wang et al., 2020) and the WFDEI forcing dataset (irradiance, Weedon et al., 2014). See SI B10 for details.

We estimated the DBH-dependent base mortality for each PFT applying a likelihood-based approach to the inventory data (SI B9). To parameterize tree growth, we focused on the carbon use efficiency (CUE = NPP/GPP) of trees under optimal growth conditions (SI B7). We modelled the CUE based on the following observations and assumptions: (1) the CUE decreases as plants grow in size, (2) the CUE under optimal conditions suffices for the observed DBH increments, (3) the CUE of trees in the inventory suffices to satisfy their respiratory needs, and (4) the order of magnitude of the CUE on stand level matches field measurements approximately (see SI B7.4).

With the modelled CUE under optimal conditions and FORMIND's submodel for primary production, we computed the GPP and NPP of trees under optimal conditions. We then used corresponding estimates of optimal DBH increments from the census data (SI B7.1) along with allometric relationships for stem dimensions to derive how much biomass trees allocate to their stem and their crown, respectively (SI B7.5). Finally, we adjusted the primary production model until enough biomass was allocated to the crowns that FORMIND's estimate of the Changbaishan forest biomass matched an estimate based on DBH-biomass relationships from the literature for each PFT (Chojnacky et al., 2014; Piponiot et al., 2022; see SI B7.5). Parameters that could not be determined via this approach were fitted so that the model best reproduced the inventory data (see below).

We assumed that trees compete for light only, but included crown defoliation as an additional process to account for the limited capacity of a forest. Trees whose GPP is insufficient to satisfy their respiratory needs loose crown biomass until all remaining parts can be maintained. Here, we assumed that – for a tree of given DBH – the maintenance respiration is proportional to the biomass. We decreased the leaf area index (LAI) of stressed trees along with their crown completeness, i.e., the ratio between current (reduced) and healthy crown biomass. Trees that have lost all their crown biomass die.

To compute the soil respiration required to determine the NEE, FORMIND uses the submodel for deadwood composition described by Sato et al. (2007), which involves a pool of fast and slowly decomposing deadwood, respectively (Paulick et al.,

2017). The corresponding decomposition rates and the transition rates between the pools are derived from the mean actual evapotranspiration (Sato et al., 2007), for which we assumed a value of $600\frac{\text{mm}}{\text{yr}}$, in line with independent estimates for the Changbaishan region (Sun et al., 2004) and earlier parameterizations of the model for temperate forests (Bohn et al., 2014).

## Model fitting

Some of the modelled processes depend on parameters not directly inferable from the available data. This included the following PFT-specific parameters: (1) the external influx of new seeds, (2) the saturation parameters of the light response curves, (3) the magnitudes of carbon losses other than maintenance respiration, and (4) the light required for seedling establishment. Furthermore, we fitted a parameter controlling the magnitude of DBH growth under optimal conditions and the sharpness of the light threshold for seedling input.

We fitted these 26 parameters using a likelihood-based approach maximizing the approximate likelihood of the inventory data, estimated from a sample of simulation results. We determined each PFT's biomass and stem count in $20\,\text{m} \times 20\,\text{m}$ forest patches. The combined information of stem count and biomass yields basic insight into the size distribution of trees: a large stem count with small biomass indicates a young forest with many small trees, and a small stem count with high biomass indicates an old forest with few large trees. Using these summary statistics instead of the full tree size distribution reduced the dimension of the considered state space, allowing us to estimate the joint distribution of the highly stochastic small-scale forest states based on a reasonable sample of simulation results. The inventory covered 625 forest patches, providing us with a similarly-sized sample of forest states.

To generate a forest state sample from the model, we first simulated $1\,\text{ha}$ of forest for a burn-in period of $2000\,\text{yr}$. Then, we sampled the forest $500$ times in $5\,\text{yr}$ intervals. We repeated this procedure $67$ times in parallel, equivalent to simulating $67\,\text{ha}$ of forest, obtaining a sample of $837,500$ forest states for each tested parameter combination.

We estimated the likelihood of the field data via kernel density estimation (KDE; Wand and Jones, 1995). In KDE, the probability density of an observation is estimated based on how many model-generated sample points are similar to the observation. Here, similarity is measured via kernel functions, which depend on bandwidth parameters. We used Gaussian kernels with bandwidths chosen corresponding to the scales of the stem counts and biomasses in the inventory data (see Table S10 in SI B11). To correct for the bias introduced when log-transforming the KDE so as to compute the log-likelihood, we applied a bias correction function derived via a first-order Taylor approximation (SI B11).

The resulting likelihood estimate converges to the true likelihood as the size of the generated sample increases and the bandwidth parameters decrease. Hence, optimizing the KDE likelihood yields consistent parameter estimates and avoids potential biases arising if the model was fitted via a deterministic modelling framework (e.g. Lehmann and Huth, 2015; Rödig et al., 2017). However, as the log-likelihood estimate is based on a sample of stochastic model results, it is stochastic as well, making it difficult to optimize. We reduced the stochasticity by decreasing the dimension of the sample space, avoiding the "curse of dimensionality" (Wand and Jones, 1995) by considering the different PFTs as mutually independent. The parameter estimates remain consistent despite this composite likelihood approach (Varin, 2008).

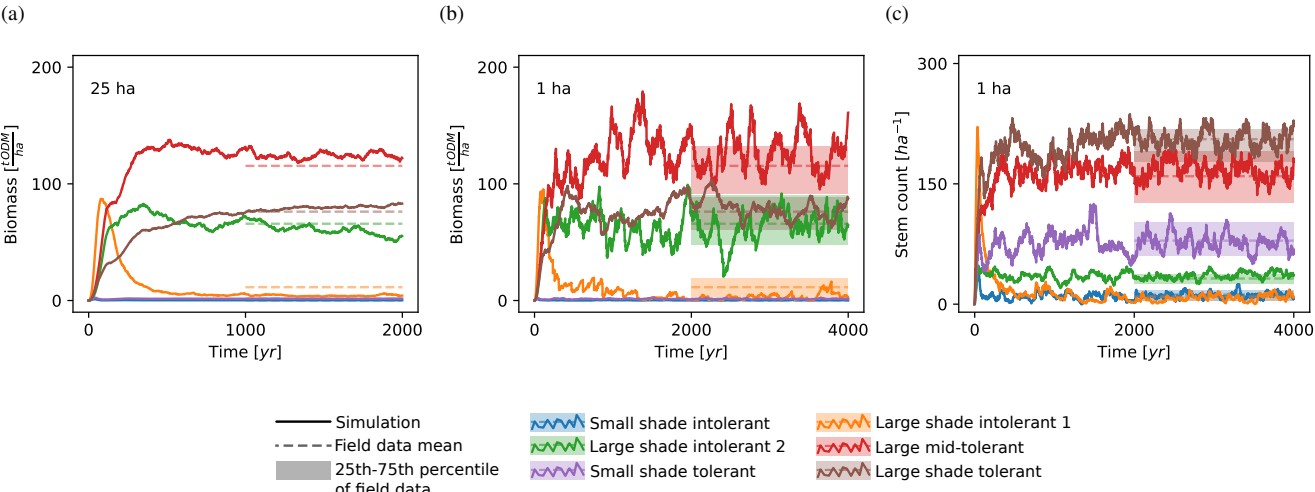

**Figure 2.** Temporal evolution of (a, b) biomass and (c) stem count of the six PFTs on (a) the $25\,\mathrm{ha}$ scale and (b, c) the $1\,\mathrm{ha}$ scale. The solid lines show the trajectory of the model simulation. For comparison, the shaded areas depict the ranges between the 25th and the 75th percentiles of the biomasses and stem counts from the inventory data. The dashed lines represent the corresponding mean values.

We maximized the likelihood by repeatedly applying a derivative-free optimization algorithm based on non-local quadratic approximations (Cartis et al., 2019). To avoid getting stuck in local minima, we used the basin-hopping algorithm (Wales and Doye, 1997), which applies multiple local optimizations with randomly perturbed initial conditions. Throughout the fitting process, we constrained the parameters to ecologically reasonable ranges. Details on model fitting can be found in SI B11. The fitted parameter values are provided in SI B.

**Size limitations**

We assumed that each tree has a maximal DBH at which it stops growing. As this maximal DBH may depend on local conditions and the tree's species, we drew the DBH limit randomly for each tree individually (details below). Trees that have reached their DBH limit are called "mature" below and are assumed to use their entire primary production for respiration.

We constructed the distributions of the DBH limits based on the maximal DBHs of the species in each PFT: for each species, we assumed that the site-dependent DBH limits are uniformly distributed between the overall maximal DBH and a value $20\%$ below this maximum. We aggregated these species-specific distributions, weighted according to the species' respective shares in the basal area of the inventory. That way, we obtained the joint distribution of DBH limits for each PFT. In SI B4.1, we describe the approach in greater detail.

### 2.3  Model validation

We validated the fitted model by visually comparing the respective marginal and joint distributions of the biomass and stem count values for the considered PFTs with the corresponding distributions observed in the field data. We created corresponding

one- and two-dimensional histograms based on both samples generated via simulations and computed based on the forest inventory data. We observed that the simulated trajectory and distribution of biomass and stem count matched the values from the inventory (Fig. 2, SI D).

To ensure the fitting algorithm did not terminate at a suboptimal local likelihood maximum, we repeated the model fitting procedure three times. We compared the resulting parameter estimates to assess how well the individual parameters are estimable. The differences between the corresponding parameter were moderate for most parameters except the light threshold for seedling establishment (SI D).

To validate the results on a broader scale ($25\,\mathrm{ha}$), we furthermore compared the modelled biomass, NPP, GPP, and LAI with values obtained for the same forest plot in independent studies (Piponiot et al., 2022). The simulated forest had a mean biomass of $270.5\,\mathrm{t\,ODM/ha}$ (estimated standard deviation for $25\,\mathrm{ha}$: $4.38\,\mathrm{t\,ODM/ha}$). Our biomass estimates from the allometric equations by Chojnacky et al. (2014) were $270.52\,\mathrm{t\,ODM/ha}$ if we only considered the major stems and $284.48\,\mathrm{t\,ODM/ha}$ for all stems in the inventory. This is below the estimate by Piponiot et al. (2022): $302\,\mathrm{t\,ODM/ha}$. The simulated forest had an aboveground wood production of $2.22\,\mathrm{t\,ODM/ha\cdot yr}$ (standard deviation: $0.07\,\mathrm{t\,ODM/ha}$; Piponiot et al., 2022: $3.55\,\mathrm{t\,ODM/ha\cdot yr}$) and GPP of $23.39\,\mathrm{t\,ODM/ha\cdot yr}$ (standard deviation: $0.2\,\mathrm{t\,ODM/ha}$; Wu et al., 2009: $29.82\text{-}33.86\,\mathrm{t\,ODM/ha\cdot yr}$). The LAI of the simulated forest was $5.18$ (standard deviation $0.05$; Liu et al., 2007: $5.08$). See SI D for details.

## 2.4   Analysis

To analyze the effect of mature trees on forest productivity, we simulated $1\,\mathrm{ha}$ of the Changbaishan forest and sampled forest characteristics and forest productivity over time on the $0.04\,\mathrm{ha}$ and the $1\,\mathrm{ha}$ scale. After a burn-in period of $2000\,\mathrm{yr}$, we analyzed the forest 1000 times in $5\,\mathrm{yr}$ time intervals. We obtained a sample of $25{,}000$ forest states on the smaller and $1{,}000$ states on the larger scale, corresponding to $1000\,\mathrm{ha}$.

To measure forest productivity, we computed the GPP, NPP, NEE, and carbon use efficiency ($\mathrm{CUE} = \mathrm{NPP/GPP}$) of the considered forest areas. We characterized the corresponding forest states by determining the basal area $A_{\mathrm{all}}$ of all trees in the forest area and the basal area $A_{\mathrm{grow}}$ of only those trees that had not reached their individual DBH limits. Based on these measures, we also determined the basal area proportion $A_{\mathrm{grow}}/A_{\mathrm{all}}$ of immature trees and the corresponding proportion of mature trees. Furthermore, we computed the DBH entropy (a measure for the diversity of DBH values; detailed explanation in section 2.5), basal-area-weighted height standard deviation, and the Shannon diversity of PFTs on the two considered scales. We weighted the plant heights by the basal areas when computing the height standard deviation so as to account for small plants having a minor impact on forest productivity.

For both considered spatial scales ($0.04\,\mathrm{ha}$ and $1\,\mathrm{ha}$), we plotted GPP, NPP, and NEE against the mentioned forest characteristics and computed the respective coefficients of determination ($R^2$) to quantify the strengths of the relationships. In a similar manner, we analyzed the relationship between the basal area proportion of mature trees and the CUE. To understand the role of the DBH entropy, we furthermore assessed its relationship with the basal area of mature and immature trees.

To assess how sensitive our results are to the assumption that mature trees stop growing completely, we computed the NPP in hypothetical scenarios in which the CUE of trees is reduced by only $50\%$, $25\%$, or $0\%$, respectively, when they enter the mature

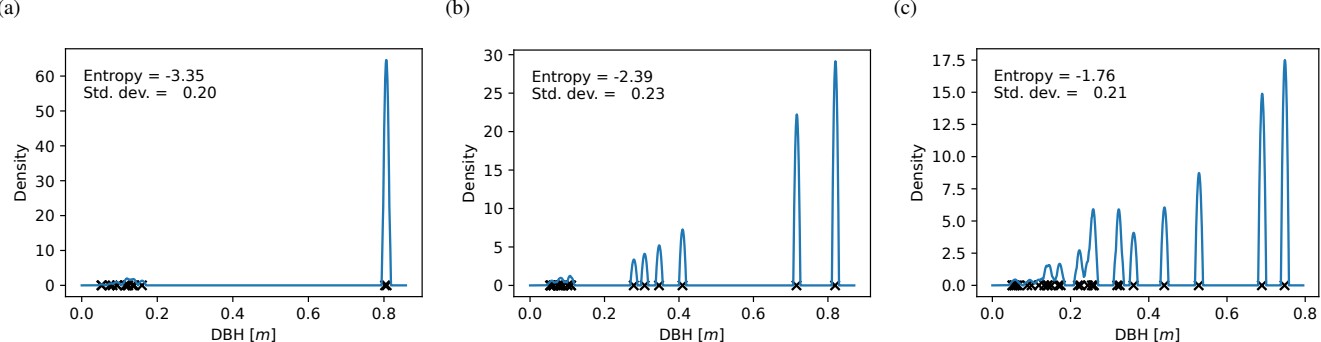

**Figure 3.** Basal-area-weighted DBH distributions for $0.04\,\text{ha}$ forest patches with (a) low, (b) intermediate, and (c) high entropy. Each black cross depicts the DBH of a plant. The height of the corresponding spike in the density function (blue line) corresponds to the plant's share in the basal area; the contributions of trees with similar DBH add up. The width of the spikes ($2h$; here: $2\text{cm}$) is the scale on which different plants are considered similarly sized. The entropy is higher the more uniformly the basal area is distributed across plants with different DBHs. In (a), two similarly large plants dominate the forest patch, whereas in (c), there are many medium-sized plants with different DBHs. Note that the standard deviation of the DBH distribution is not related to the DBH entropy.

stage. To avoid refitting the model for each of these validation scenarios, we adjusted only the intrinsic carbon fluxes and held the sizes of the mature trees constant. We then analyzed the relationship of the obtained NPP values with the covariates given above.

### 2.5 DBH entropy as a proxy for the prevalence of mature trees

It is difficult to know which trees have reached their site-dependent growth limits in field studies. Hence, a proxy for the prevalence of mature trees is needed in practice. Such a proxy should be easy to compute from inventory data and may account for the following working hypotheses: (1) forest patches dominated by mature trees consist of a small number of large individuals preventing the existence of medium-sized trees; (2) in old-growth forests, individuals typically differ in age and size, but mature individuals of the same species may have similar DBH values. The proxy should also reflect that large trees have a higher impact on forest dynamics than small trees.

As a proxy satisfying these requirements, we propose the basal-area-weighted DBH entropy $S_{\text{DBH}}$ (below simply "DBH entropy"), defined as the entropy of the distribution of DBHs in a forest patch (cf. Staudhammer and LeMay, 2001; Park et al., 2019). If we split the range of occurring DBH values into equally sized intervals $I$ and determined the basal area share $p_I$ of trees in each size class $I$ relative to the total total basal area, the DBH entropy could be approximated via

$$S_{\text{DBH}} = -\sum_{I \in \mathcal{I}} p_I \ln(p_I). \tag{1}$$

Here, $\mathcal{I}$ is the set of DBH classes and

$$p_I = \frac{\sum_{d \in I} d^2}{\sum_{I \in \mathcal{I}} \sum_{d \in I} d^2} \tag{2}$$

is the basal area share of trees in size class $I$.

The weights $p_I$ can be interpreted as probabilities indicating how likely we would obtain a tree from size class $I$ if we randomly selected trees from the forest patch with probabilities proportional to their basal areas. The entropy is higher the more evenly the the DBHs are distributed (Fig. 3). If the forest patch is dominated by one or a few large trees, it is likely that we draw one of their size classes, making the entropy small. Similarly, if two trees have a similar DBH, the probability to pick a tree from their size class increases, decreasing the entropy. Since we weight the DBH distribution by the basal areas, adding small trees to the forest patch does not change the entropy significantly.

As the approach presented above is sensitive to the specific choice of interval bounds, we used a more robust definition of the DBH entropy in our analysis (SI C1). We applied kernel smoothing (Wand and Jones, 1995) with an Epanechnikov kernel to obtain a continuous estimate of the DBH distribution instead of discrete probabilities $p_I$ (cf. Fig. 3), and we exchanged the sum in equation (1) with an integral. Kernel smoothing requires a bandwidth parameter (here: 1cm), which is comparable to the width of the DBH intervals $I$ and defines the scale on which two trees are regarded similar.

## 3 Results

The basal area of the forest stand was strongly correlated with the GPP, irrespective of the spatial scale ($R^2 \geq 0.65$; Figs. 4a and 5a). For the NEE, these correlations were much weaker ($R^2 \leq 0.1$; Figs. 4f, 5f) and for the NPP merely existent ($R^2 = 0$; Figs. 4k, 5k). This contrasts with the basal area of immature trees: here, the correlations were small for the GPP ($R^2 \leq 0.15$; Figs. 4b, 5b) but large for the NPP ($R^2 \geq 0.74$; Figs. 4g, 5g) and the NEE ($R^2 \geq 0.59$; Figs. 4l, 5l). We obtained a similar but slightly weaker result for the DBH entropy. On the small scale ($0.04\,\mathrm{ha}$), it was weakly correlated with the GPP ($R^2 = 0.11$; Fig. 4c) but strongly correlated with NPP ($R^2 = 0.47$; Fig. 4h) and NEE ($R^2 = 0.39$; Fig. 4m). These correlations decreased on the larger scale ($1\,\mathrm{ha}$; $R^2 \leq 0.26$; Figs. 5c, h, m).

The weighted tree height standard deviation was strongly negatively correlated with the GPP ($R^2 \geq 0.56$; Figs. 4d, 5d) but almost uncorrelated with NPP and NEE ($\left|R^2\right| \leq 0.03$; Figs. 4/5i, n) on both spatial scales. The Shannon diversity of PFTs was moderately correlated with the NPP ($R^2 \in [0.17, 0.19]$; Figs. 4j, 5j), weakly correlated with the NEE ($R^2 \leq 0.04$; Figs. 4o, 5o), and weakly negatively correlated with the GPP ($R^2 \leq 0.07$; Figs. 4e, 5e).

The DBH entropy was positively correlated to the basal area of immature trees ($R^2 = 0.33$ on the small scale; Fig. 6a) and weakly negatively correlated to the basal area of mature trees ($R^2 = 0.09$; Fig. 6c). For the latter, the DBH entropy was a poor predictor in forest patches with large overall basal area. On the hectare scale, the relationships became weaker for immature trees ($R^2 = 0.23$; Fig. 6b) but stronger for mature trees ($R^2 = 0.19$; Fig. 6d).

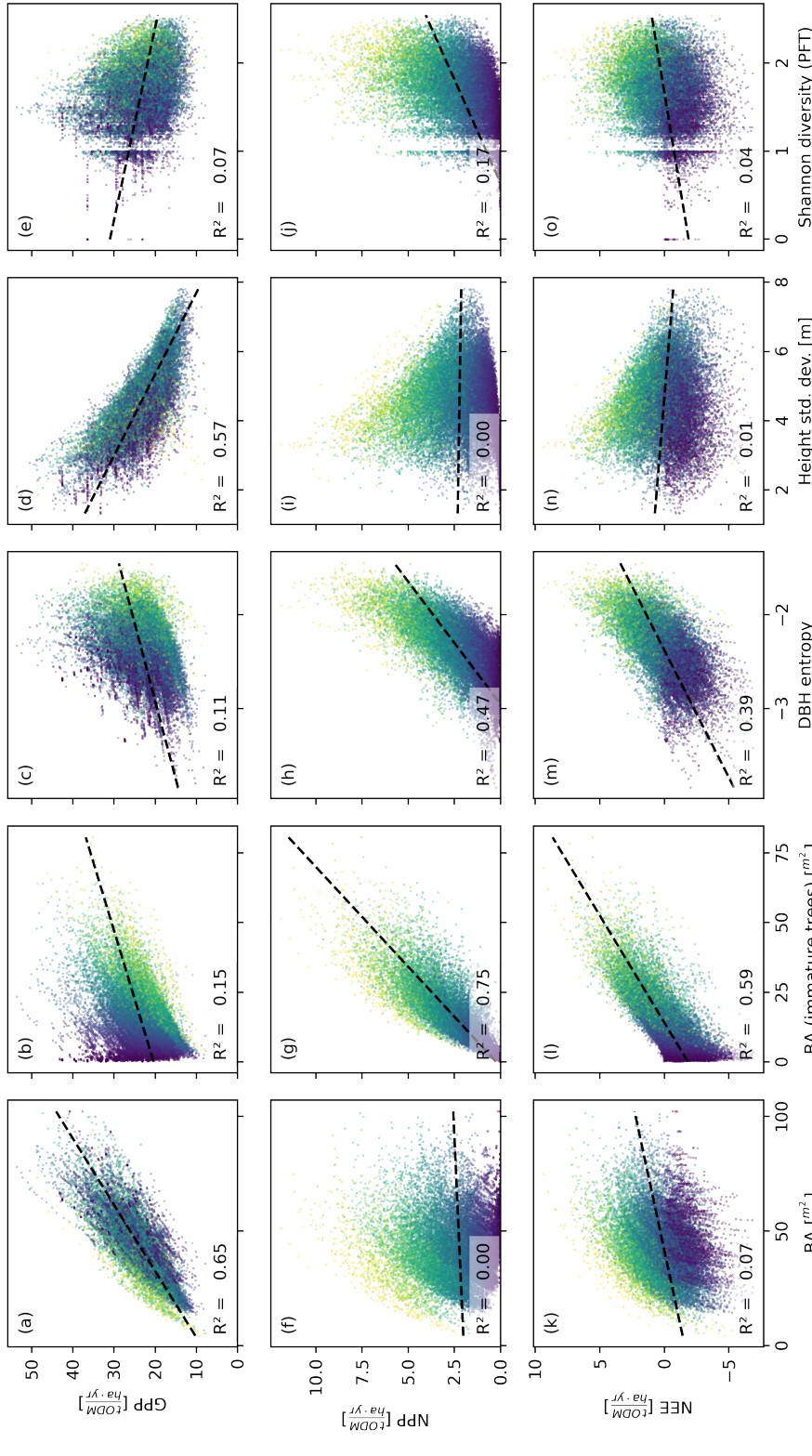

**Figure 4.** Productivity measures (GPP, NPP, and NEE) dependent on different measures of basal area (BA) and heterogeneity. Each dot corresponds to a 0.04ha forest patch (sample size: 25, 000). The colour indicates the basal area proportion of mature trees (blue: only mature trees; yellow: no mature trees). The GPP is mainly driven by the basal area, whereas NPP and NEE are driven by the basal area of immature trees. The heterogeneity measures are generally poorer predictors than the basal area measures. Among the former, the DBH entropy has the best predictive capacity for NPP and NEE and may serve as a valuable proxy if distinguishing mature and immature trees is not possible.

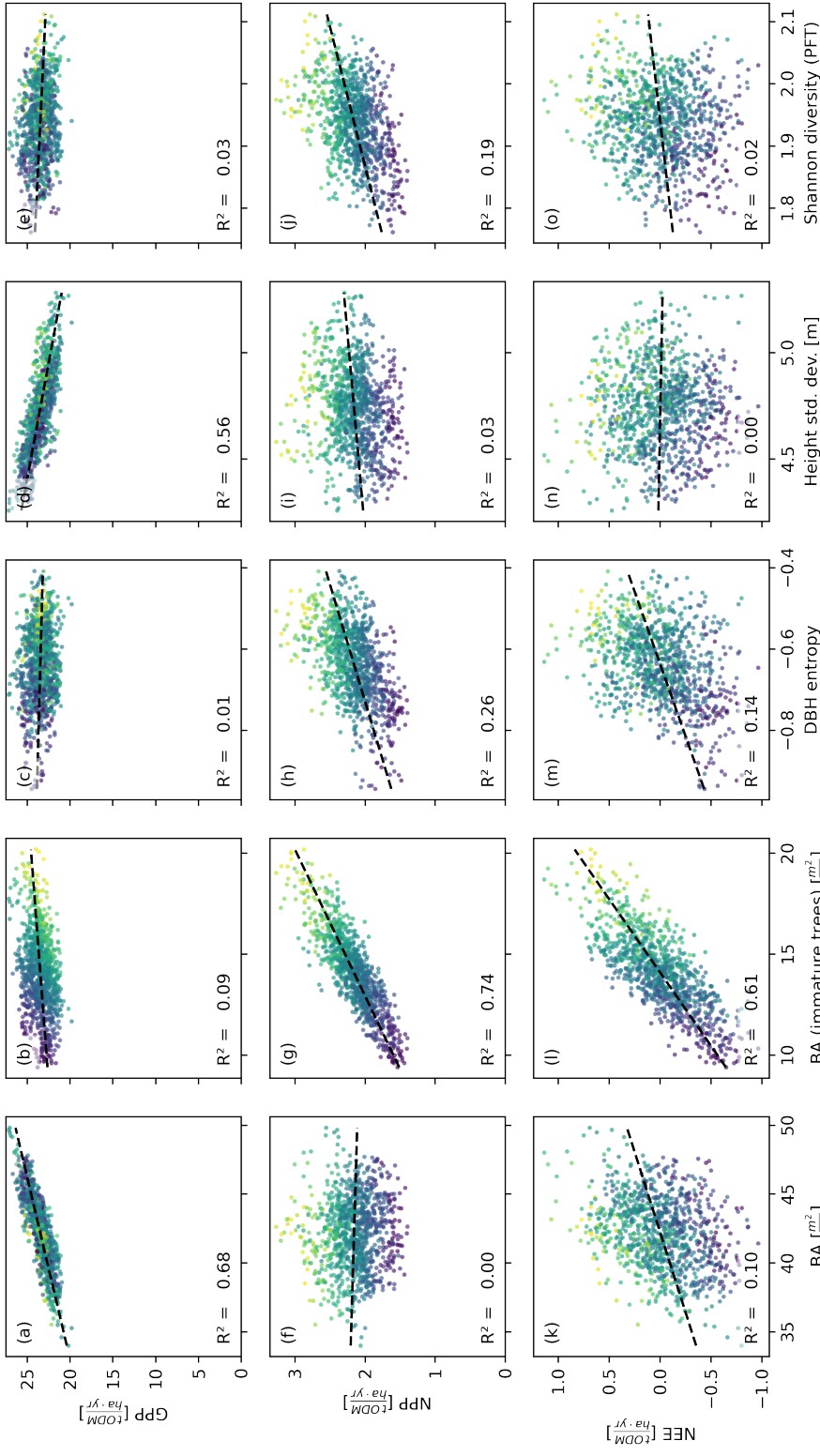

**Figure 5.** Productivity measures dependent on different measures for basal area and heterogeneity. Each dot corresponds to a 1 ha forest patch (sample size: 1,000). The colour indicates the basal area proportion of mature trees (blue: only mature trees; yellow: no mature trees). The correlation patterns resemble those observed on the finer scale (Fig. 4). Only the DBH entropy looses predictive power.

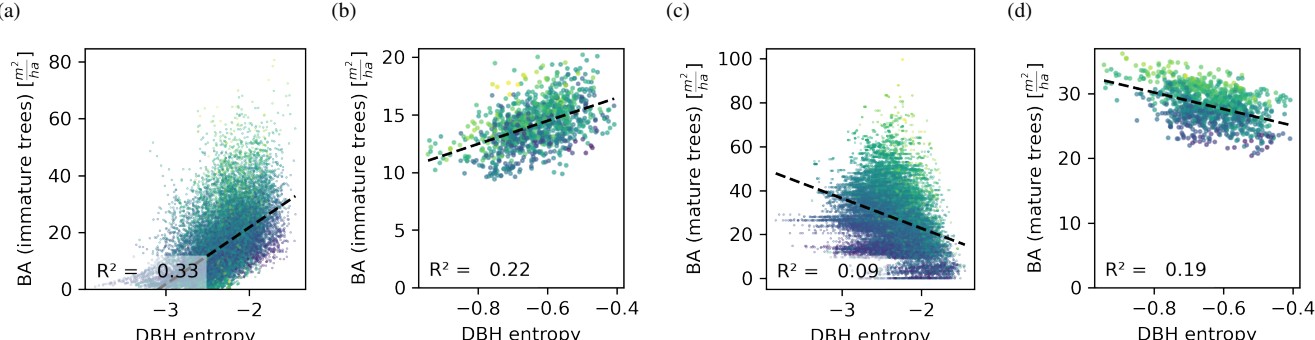

**Figure 6.** Relationship between the DBH entropy and (a, b) the basal area of immature and (c, d) mature trees, depicted on (a, c) the $0.04\,\mathrm{ha}$ and (b, d) the $1\,\mathrm{ha}$ scale. Each dot corresponds to a forest patch of the respective scale. The colour corresponds to the total basal area (dark: low, light: high). The DBH entropy correlates positively with the basal area of immature trees, which drive the NPP, and correlates negatively with the basal area of mature trees, which do not contribute to the NPP and compete with immature trees. The relationships are stronger on the small scale.

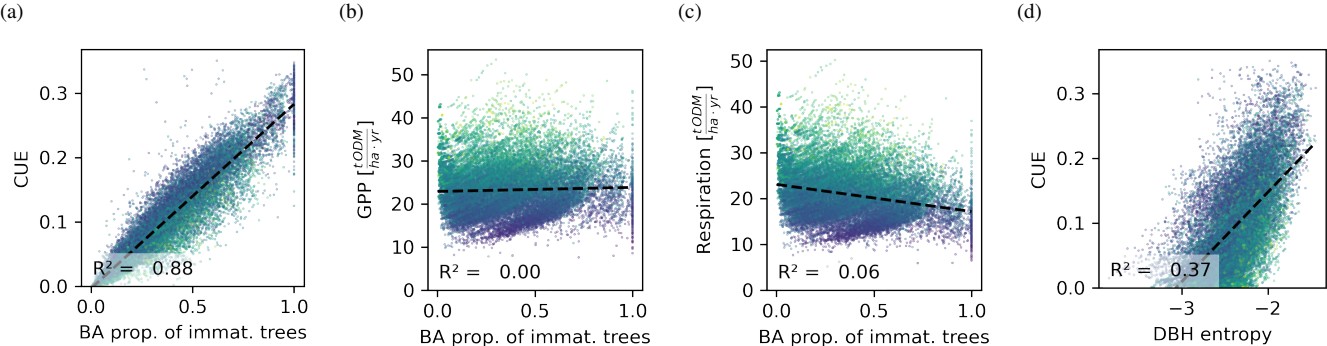

**Figure 7.** Relationship between the basal area proportion of immature trees and (a) the CUE, (b) the GPP, and (c) the tree respiration / carbon losses. The CUE is proportional to the basal area of immature trees. Though the CUE can be directly computed from the GPP and respiration, a similar relationship is not visible for these, indicating that they are not the drivers behind the proportionality. (d) Relationship between the DBH entropy and the CUE. Though this relationship is weaker than that between the proportion of immature trees and and CUE, the DBH entropy may serve as a proxy for the CUE.

Each dot corresponds to a $0.04\,\mathrm{ha}$ forest patch. The colour corresponds to the basal area (dark: low, light: high).

The CUE was proportional to the proportion of immature trees in the forest (Fig. 7a). The regression analysis yielded an intercept of $3.16 \cdot 10^{-3}$ on the small and $2.182 \cdot 10^{-4}$ on the large scale, with $R^2$ values of $0.88$ and $0.82$, respectively. The proportionality constants (slopes of the fitted curves) were $0.29$ and $0.28$. The relationship between CUE and DBH entropy was also significant, but weaker ($R^2 = 0.3$ on the small and $R^2 = 0.28$ on the large scale; Fig. 7). The GPP and tree respiration / carbon losses did not show a strong correlation with the proportion of immature trees ($R^2 = 0$ and $R^2 = 006$, respectively on the small scale; Figs. 7b, c).

Decreasing the CUE reduction of mature trees in comparison to similar immature trees decreased the correlation between the basal area of immature trees and the NPP. However, the predictive performance remained high ($R^2 \geq 0.53$) even if the CUE of mature trees was only reduced by $25\%$ (Fig. S1). The DBH entropy was even less sensitive to a change in the CUE reduction. However, when the CUE was reduced by less than $50\%$, the DBH entropy computed with cubic DBH weights had a stronger correlation with NPP than the basal-area-weighted version and achieved an even higher $R^2$ of $0.57$ and $0.55$ when the CUE was reduced by only $25\%$ and $0\%$, respectively. Details and further results regarding the CUE reduction scenarios are presented in SI E.

## 4   Discussion

We suggested a simple framework of "mature" and "immature" trees to disentangle the impact of competition and intrinsic growth limitations on forest productivity in old-growth forests. Thereby, we found that the drivers of NPP and NEE were distinct from those determining GPP. While the latter was strongly correlated with the total basal area, NPP and NEE were related to the basal area of immature trees only. This indicates that the increased respiratory losses of mature trees play the major role in forests' carbon balance: despite having a significant GPP, mature trees contribute less to wood production but rather reduce the productivity of other trees via competition. Hence, tree maturity may be a major driver of the difference between NPP and GPP, making GPP-related covariates such as light competition insufficient to explain local variations in NPP and NEE.

This conclusion is supported by the observed proportionality between the CUE and the basal area share of immature trees: carbon usage was more efficient the more the forest was dominated by immature trees. The proportionality can be explained by noting that (1) the individual-level GPP is strongly positively correlated with basal area, irrespective of the maturity stage, and (2) only immature trees contribute to the NPP. On the stand level, neither the GPP nor the respiration were correlated with the proportion of immature trees, showing that the proportionality was not driven by the decreased GPP or increased respiration of forests with a high share of mature trees.

These findings are based on a maturity definition considering the individual trees' growth potential in the absence of competition. This potential can be challenging to determine in field studies, as it requires to identify the causes of individuals' growth limitations. Hence, alternative maturity definitions, based on tree size or signs of senescence may be used (Gibbons et al., 2008). Applying such alternative maturity definitions will yield qualitatively similar results if the considered characteristics are strongly correlated with the trees' growth potential. Otherwise, other structural forest attributes may be considered.

To that end, we suggested the DBH entropy as a proxy for the prevalence of immature trees and thereby NPP and NEE. The DBH entropy was positively correlated with the basal area of immature trees and negatively correlated with the basal area of mature trees, but its relationships to NPP and NEE were even stronger. This indicates that the predictive capacity of the DBH entropy stems not only from its correlation with the prevalence of immature trees but also from other mechanisms. In line with this observation, the DBH entropy remained a good predictor for NPP even in the validation scenario where we did not reduce the CUE of mature trees (SI E). These findings support previous studies identifying structural diversity as a major driver of forest productivity (Dănescu et al., 2016; Bohn and Huth, 2017; Silva Pedro et al., 2017; Bohn et al., 2018; Park et al., 2019; LaRue et al., 2023). Note that our DBH entropy index differs from the classic entropy-based measures for structural diversity (Staudhammer and LeMay, 2001) by the basal-area-based weighting (Park et al., 2019), which improved its predictive power (SI C2).

Remarkably, the height standard deviation, another measure for structural diversity, did not have a significant positive correlation to any of the productivity measures. The height standard deviation depends on the width of the height spectrum, i.e., the difference between the height of the smallest and the largest tree. Hence, forests can exhibit a high standard deviation even if their diversity of tree heights is low. This contrasts with the entropy, which measures how many different tree sizes there are without regarding their actual values. The strong negative relationship between the height standard deviation and GPP can be explained by the weighting we applied. Weighting the tree heights by basal area decreases the standard deviation in forest stands with many large trees, which in turn have a large GPP.

The Shannon diversity of PFTs was not strongly related to any of the forest productivity measures. This was due to the differences between stem count and biomass of the PFTs. Four PFTs contributed significantly to the forest's stem count and thus the Shannon diversity. In contrast, the biomass was dominated by two PFTs only, which consequently contributed most to the production. Hence, the Shannon diversity of PFTs was a poor predictor for productivity. However, if the Shannon diversity was computed based on tree species rather than PFTs, it could yield useful information on the diversity of the DBH limits, as these are species dependent. Setting this diversity of limits into relation with the actual diversity (or entropy) of DBH values could hence improve NPP estimates.

Changing the spatial scale from $0.04\,\mathrm{ha}$ to $1\,\mathrm{ha}$ did not alter most of the relationships we considered. By construction, the coefficient of determination is insensitive to the addition of independently identically distributed random variables. As the interactions between forest patches were weak and the basal area, GPP, NPP, and NEE are additive measures, their respective correlations were not affected by the scale. The same applied to the height standard deviation, which is additive if the weighted mean height is approximately constant in all small-scale patches. The Shannon diversity of PFTs did not show strong patterns on any scale. The DBH entropy, however, was most informative on a small scale (e.g. $0.04\,\mathrm{ha}$). On large scales (e.g. $1\,\mathrm{ha}$), the entropy increases and varies less between forest sections, since more trees are considered. This is a significant finding, as many previous studies considered entropy-based diversity indices on larger scales (often $\geq 0.5\,\mathrm{ha}$; Dănescu et al. 2016; Silva Pedro et al. 2017; Park et al. 2019). In line with our results, a loss of information on larger scales was noticed by Chisholm et al. (2013) with respect to the Shannon index. Nonetheless, if the scale is smaller than that of plant interactions, the DBH entropy cannot reflect information on competition and dominance, and the similarities between mature trees cannot be reflected.

## 4.1 Model parameterization and limitations

Being an individual- and process-based model, FORMIND is designed to attain high mechanistic realism while achieving the computational performance required to study forest dynamics on large spatial and temporal scales (Fischer et al., 2016). Hence, some processes such as plant-internal signalling, dynamics of nonstructural carbon, below-ground carbon dynamics, interactions with mycorrhiza, or pest-induced stress are not covered explicitly but implicitly incorporated into high-level processes. As a result, not all aspects of forest community dynamics may be reproducible with the model. Nonetheless, the main carbon fluxes are covered, allowing us to differentiate immature trees from those that have reached their maximal sizes and to analyze carbon fluxes on small spatial scales.

Measuring GPP and NEE on small scales is challenging, since eddy covariance measurements, for example, typically apply to the whole stand level only, are costly and bound to one location due to the immobility of the measurement towers. The model-based approach required some innovations in model design and parameter estimation. For example, the likelihood-based fitting method allowed us to estimate parameters based on small-scale (here: $0.04 \, \text{ha}$) forest characteristics despite their stochastic variations. The small-scale distribution of stem counts and biomass contains information on local interactions and consequently the range and diversity of local states a forest can attain. This information is typically lost on larger scales. Circumventing the need to reduce stochasticity via aggregation over several hectares of forest (see e.g. Rödig et al., 2017) enabled us to estimate parameters affecting the small-scale forest dynamics and allowed us optimize 26 parameters on regeneration, light response, optimal growth, and respiration. Applying a parameterization framework focusing on the tree-level carbon use efficiency guaranteed a balanced parameterization of the individual-level NPP and GPP.

Our fitting approach also circumvented challenges typically arising in the Bayesian framework. Bayesian methods, such as approximate Bayesian computation (ABC; Beaumont et al., 2002; Csilléry et al., 2010), require the evaluation of many parameter combinations. This is computationally costly in models for old-growth forests, as the entire succession has to be simulated. Furthermore, the stochastic search performed in ABC and classical Markov Chain Monte Carlo methods may fail to find good parameter combinations when the parameter space is large. Hence, our methodological advances can also benefit future forest models.

The good match between the biomass and stem count distributions in the simulated forest and the inventory indicates that the model replicates the forest structure well. Further validation via independent estimates of biomass, GPP, NPP, and LAI showed that the model reproduces major forest dynamics. Nonetheless, the model underestimated the mean biomass, GPP, and NPP. The underestimated biomass resulted partially from our focus on the main stems in the inventory, neglecting additional minor stems. Including the secondary stems as separate trees would have led to overestimated LAI values, causing forest thinning and making it difficult to fit the dynamic model to the field data. The biomass bias, along with the assumption that mature trees stop growing, may also have caused the underestimated NPP and GPP. Nonetheless, these quantitative differences do not invalidate the strong qualitative results we obtained.

The strong correlation we observed between basal area and GPP may stem from our assumption that leaf area and basal area are proportional within a PFT. Though this assumption is in line with theoretical and empirical findings (West et al., 1999; Xu

et al., 2021), local conditions and competition can blur this relationship in practice, weakening it in field observations. As an alternative, the GPP could be estimated from stand-level LAI values (see e.g. Xie et al., 2019).

The relationship between basal area and GPP could also be weakened by competition for water and other resources, which might also yield other interactions between mature and immature trees. Added competition may strengthen the negative effect of mature trees on forest productivity. As a result, the relationship between the proportion of immature trees and the CUE would become non-linear, with a disproportionally low CUE in stands dominated by mature trees. Consequently, the basal area of mature trees would need to be considered in addition to the basal area of immature trees to accurately estimate NPP and

NEE.

    In special cases, mature trees could also have positive effects on smaller trees, for example by providing shelter (Lett and Dorrepaal, 2018) and improving soil conditions (Yunusa and Newton, 2003). In forests whose dynamics are driven by sink limitations (i.e., limitations affecting carbon allocation to growth) rather than source limitations (limitations affecting carbon supply), such effects could induce a positive effect of mature trees on NPP.

Our analysis built on the assumption that trees have maximal sizes. We modelled this via an abrupt transition from the growing to the mature stage, which is a common approach in forest modelling (Shugart et al., 2018). In reality this transition can be gradual, and trees may require minimal DBH increments to maintain the function of their vascular system (Prislan et al., 2013). However, our results remained consistent even if the CUE of mature trees was only reduced by $25\%$ as compared to immature trees of the same size, suggesting that life-stage-dependent carbon losses have a dominant impact on the forest

dynamics even if they have a moderate magnitude. Though the concept of growth limitations acting on the individual scale is subject to an ongoing debate (Stephenson et al., 2014; Foster et al., 2016; Sheil et al., 2017; Forrester, 2021; Anderson-Teixeira et al., 2022), there is strong evidence that the NPP and / or CUE decrease with the age of forest stands (Gower et al., 1996; Tang et al., 2014; Collalti et al., 2020a), indicating that tree age or size have a significant effect on individual biomass increment (West, 2020).

We considered a forest under spatially and temporally uniform environmental conditions to study the within-stand productivity variations and their connection with forest structure. Temporal climatic variations and changing occurrence of diseases and pests could increase the variance of GPP, NPP, and NEE, and weaken their correlations with forest attributes. Though the external factors could become a major driver of forest dynamics, temporal averaging could reduce the resulting productivity variations so that data obtained on longer time scales might show patterns similar to those presented here. This could be

confirmed via further simulation studies, e.g. with an extended FORMIND parameterization incorporating variable climate.

    Similarly, spatial heterogeneity in climate, soil, species composition, and other factors could affect forest productivity on larger scales (Munné-Bosch, 2018; West, 2020; Gea-Izquierdo and Sánchez-González, 2022). To appropriately account for these variations, our results would need to be combined with appropriate stand-level covariates to obtain productivity estimates on regional scales. Nevertheless, our findings may be applicable to extended areas with comparable environmental conditions.

## 4.2 Outlook

Using the concept of the potential CUE to characterize tree maturity could become a useful framework to understand forest productivity on local scales. The identification of mature trees, whose growth is primarily limited by intrinsic factors, may be conducted irrespective of the mechanism behind the limitations, be it increased respiratory losses (O'Leary et al., 2019), sink limitations (Potkay et al., 2022), limited nutrient or water availability (Munné-Bosch, 2018), or even genetic predisposition (Liu et al., 2016). As we used a generic forest model and our results were robust across scales, our observations may hint towards a universal relationship between tree maturity and forest productivity. This connection could be used to develop new theory that could eventually lead to accurate predictions of NPP and NEE based on general forest characteristics. Such predictions have proven difficult in the past (Chisholm et al., 2013; Rödig et al., 2018) but could be highly relevant for a broad spectrum of applied and theoretical questions in forest ecosystem science. Here, the DBH entropy could prove particularly useful, as it can be easily obtained from inventory data and may serve both as a measure for forests' structural diversity on the local scale and as a proxy for net forest productivity in old-growth forests.

Confirming and generalizing the observed relationships between tree maturity, DBH entropy, NPP, and NEE is a promising endeavour for both theoretical and field studies. Further modelling studies could assess the expected strength of the relationships in forests in different successional stages, under varying environmental conditions, and in the presence of additional stressors such as competition for nutrients and water. Field studies could attempt to validate these findings. Typical DBH maxima are documented for many species from temperate forests and could serve as a first proxy for maturity (Aiba and Kohyama, 1997; Kohyama et al., 2003; Russell and Weiskittel, 2011; del Río et al., 2019). Combining the gained insights with large-scale predictors for forest productivity could then lead to a unified theory of forest productivity.

## 5 Conclusions

We applied a modelling approach to investigate how the prevalence of mature (full-grown) trees and forest structure explain within-stand variations of forest productivity. We found that NPP and NEE are mainly driven by the basal area of immature trees, whereas the GPP depends on the total basal area. This suggests that loss-induced limitations rather than variations in GPP determine NPP and NEE.

The forest stand CUE was proportional to the basal area share of immature trees. We suggested and tested the basal-area-weighted DBH entropy as an easy-to-compute proxy for both the prevalence of mature trees and NPP and NEE. Other measures for structural diversity, namely the height standard deviation and the Shannon entropy of functional types, had much smaller predictive power. Our results were robust across spatial scales, and due to their solid mechanistic foundation and our generic model, our findings yield promising hypotheses for field studies and new theoretical work. For example, we hypothesize that within-stand NPP increases with the DBH entropy and that focusing on immature (not full-grown) trees could yield more accurate structure-productivity relationships.

*Author contributions.* SMF and AH jointly conceived the study. XW contributed the field data. SMF parameterized the model with substantial input by AH and conducted the data analysis. SMF and AH jointly conceived the manuscript; SMF wrote the manuscript; AH revised the manuscript. All authors approved the manuscript.

*Competing interests.* The authors declare no competing interest.

*Acknowledgements.* The authors would like to thank the members of the vegetation modelling group at the UFZ and two anonymous reviewers for helpful discussions and feedback. This research was conducted as part of the project "The role of species traits and forest structure on spatial carbon dynamics of temperate forests" (ForCTrait), established within the cooperation "China-NSFC-DFG 2019" between the Deutsche Forschungsgemeinschaft (DFG, German Research Foundation) and the Natural Science Foundation of China (NSFC). This work was funded by the Deutsche Forschungsgemeinschaft (DFG) – 43150473.

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
