# Peer review of "Distinguishing mature and immature trees allows to estimate forest carbon uptake from stand structure"

_EGUsphere, 2023_

## Referee Comment (RC1)

General Comments:

The manuscript applied the existing individual-based forest gap model (FORMIND), which was developed using data from an old-growth temperate forest in the northeastern China. Authors present a novel methodology to distinguish between mature and immature trees to understand forest productivity. This approach offers a fresh perspective compared to traditional methods focusing mainly on gross primary production (GPP). The manuscript is well-organized, systematically presenting its research approach, results, and conclusions.

Manuscript is interesting and useful to international audience of the journal. However, there is room for improvement in the manuscript. The approach and conclusions are somewhat limited by the methodological framework and the absence of a comprehensive analysis of the ecological implications. Authors are suggested to address the following issues while making the revision.

- The introduction provides adequate background but lacks a critical review of previous research methodologies (other process-based models) and their limitations. Also, there is a need to introduce the limitation of the current FORMIND model. Suggest to enhance the introduction.
- Different species might exhibit significant variations in growth and carbon dynamics, even within the same maturity classification. The selection and classification of trees into mature and immature categories are not sufficiently justified. More rigorous criteria and a discussion of potential biases in these classifications are needed.
- The choice of the FORMIND model may not fully capture the complexity of forest dynamics, especially in terms of species-specific interactions and responses to environmental variables. For instance, you did not apply any space competition in the model. Suggest to compare results with those obtained from alternative models, particularly those incorporating more detailed species-specific parameters or interactions with abiotic factors.
- The acknowledgment of the model's limitations is a positive aspect, but the discussion lacks a critical assessment of how these limitations might have influenced the study's conclusions. Suggestions for alternative modeling approaches or supplementary methods to address these limitations would provide a more balanced view.
- A lot of supplement information is provided with the manuscript. I'm not sure if it can refer to other fundamental literature previously published. Are the allometric relationships part of the FORMIND model? It would be better to keep concise and easier for the readers to understand.

Detailed comments:

1. Introduction: Consider providing a brief introduction on any challenges or limitations encountered while adapting the FORMIND model to this specific old-growth temperate forest.

2. Introduction: P2 second paragraph: "Nonetheless, it has proven difficult to identify clear relationships between forest structure and NPP (Chisholm et al., 2013) as several factors interact...": Suggest elaborating on the specific factors that complicate the relationship between forest structure and NPP.

3. Method: Page 4 2.1 Field data: I have concerns about the allometry information and biomass equations provided in Supplementary A. (1) A lot of species lack allometry data and biomass equations. How did you address these species in your study? Did you use likelihood-based

analysis similar to the PFTs classification? Please clarify this in the methods part. (2) The biomass equations, adopted from Chojnacky et al. (2014), are generalized primarily for North American species. Since most of the equations are empirical models, I doubt their accuracy when used directly.

4. Methods: Page 6 Model fitting, second paragraph, "We fitted these 18 parameters...": cannot get 18 parameters based on your description, please clarify.

5. Results, first paragraph, "...This contrasts with the basal area of immature trees." please add supporting figures or statistical results.

6. Results: "...These results are depicted in Figures 4 and 5." Figures should be accompanied by the corresponding results in brackets. This will make it clearer for readers to correlate the interpretation with the figures.

7. Results: "...These results are shown in Fig. 6." Similar suggestions as above.

8. Results, Figure 7, does each dot correspond to a forest patch of 0.04ha? Please clarify.

9. Discussion, Page15, second paragraph, "The proportionality can be explained by the strong connection between the individual-level basal area and GPP in conjunction with the negligible NPP of mature trees." Consider simplifying or re-framing it for better readability.

10. Discussion, Page 15, second paragraph, "On the stand level, however, neither the GPP nor the respiration were correlated with the proportion of immature trees (Fig. 7) ...", in the discussion section, only this sentence refers to the figures. Please maintain consistency.

---

## Author Comment (AC2)

**General Comments:**

The manuscript applied the existing individual-based forest gap model (FORMIND), which was developed using data from an old-growth temperate forest in the northeastern China. Authors present a novel methodology to distinguish between mature and immature trees to understand forest productivity. This approach offers a fresh perspective compared to traditional methods focusing mainly on gross primary production (GPP). The manuscript is well-organized, systematically presenting its research approach, results, and conclusions.

Manuscript is interesting and useful to international audience of the journal. However, there is room for improvement in the manuscript. The approach and conclusions are somewhat limited by the methodological framework and the absence of a comprehensive analysis of the ecological implications. Authors are suggested to address the following issues while making the revision.

> Thank you for your thorough review of our manuscript and the many helpful comments. Below, we respond to each of them and explain how we would like to address them in a potential revision.

> Our responses are written in blue font and indented.

The introduction provides adequate background but lacks a critical review of previous research methodologies (other process-based models) and their limitations. Also, there is a need to introduce the limitation of the current FORMIND model. Suggest to enhance the introduction.

> Thank you for this suggestion. We will add a short review of other process-based models to the 2nd paragraph after the research questions on page 3.

> Of course, FORMIND and other forest models have limitations, which are very important to understand to correctly interpret the results. In order to appropriately account for these limitations, it is important to be specific and explain how each individual limitation may affect the results. Otherwise, we would be at risk of leaving the reader with common sense statements about the limitations of modelling in general.

> We believe that the introduction, which we intend to set the frame of the study and to provide general guidance to the reader, is not the best place for a discussion of limitations with the necessary level of detail. Therefore, we provided a thorough discussion of limitations in section 4.1. We will extend this section further in response to your comments below.

Different species might exhibit significant variations in growth and carbon dynamics, even within the same maturity classification. The selection and classification of trees into mature and immature categories are not sufficiently justified. More rigorous criteria and a discussion of potential biases in these classifications are needed.

> Thank you for this thoughtful observation. We agree that a binary distinction of mature and immature trees could be difficult in field studies (as acknowledged in section 4.1; 2nd paragraph on p. 18).

> Our maturity definition, based on the carbon use efficiency (CUE) that an individual could attain in the absence of competition, allows us to distinguish competition (which is production-related) and individual growth bounds (loss-related) as main limiting drivers of NPP. We admit that this may not have been sufficiently clear in the paper. We will therefore

clarify and adjust the respective sections in the introduction and discussion accordingly. Furthermore, we will add a discussion of potential biases to the text.

For a potential application in field studies, we suggested and analyzed the DBH entropy as a proxy for tree maturity. However, within the context of our modelling study, we could also identify the process (i.e., individual limitations rather than competition) that drives productivity.

To account for the species- and individual-specific differences in carbon dynamics, we incorporated species-dependent and randomized growth limitations into the model.

The choice of the FORMIND model may not fully capture the complexity of forest dynamics, especially in terms of species-specific interactions and responses to environmental variables. For instance, you did not apply any space competition in the model. Suggest to compare results with those obtained from alternative models, particularly those incorporating more detailed species-specific parameters or interactions with abiotic factors.

Thank you for this suggestion. The model *does* feature space competition, but we may have caused some confusion by misleadingly using the word "space competition" for "crowding mortality", which is typically modelled as increased (stochastic) tree mortality in "full" forest patches. We did not use this stochastic phenomenological model for space competition. Instead, as a new innovation, we took a more mechanistic approach and introduced tree mortality due to strong light competition. That is, trees that do not receive enough light to fulfill their respiratory needs die.

This process has the same effect as crowding mortality ("full" forests lead to deadly overshadowing of small plants) but has a better mechanistic justification. In fact, both the species of the shadowing and the overshadowed trees are considered explicitly, since the LAI of larger plants as well as the respiratory demands of smaller plants are parameterized individually for each plant functional type (PFT). We will clarify this in the supplement section B.8 and change the wording of "space competition".

The acknowledgment of the model's limitations is a positive aspect, but the discussion lacks a critical assessment of how these limitations might have influenced the study's conclusions. Suggestions for alternative modeling approaches or supplementary methods to address these limitations would provide a more balanced view.

Thank you for pointing this out. We will extend the limitation section 4.1 to cover the effect of the limitations on the results more explicitly.

A lot of supplement information is provided with the manuscript. I'm not sure if it can refer to other fundamental literature previously published. Are the allometric relationships part of the FORMIND model? It would be better to keep concise and easier for the readers to understand.

Thank you for this question and comment. Our study contains several innovations that were necessary to accurately reproduce the behaviour of the Changbaishan forest. The increased accuracy came at the cost of a longer paper supplement. The supplement also contains the other values of the Changbaishan parameterization. This is required for full reproducibility of the study.

We understand that the length of the supplement poses a challenge to readers, who typically need to quickly identify the information most relevant to them. We will therefore provide a

table of contents and summary of our model extensions at the beginning of the supplement to guide the reader.

**Detailed comments:**

Introduction: Consider providing a brief introduction on any challenges or limitations encountered while adapting the FORMIND model to this specific old-growth temperate forest.

Thank you for this suggestion. As the parameterization is not the focus of our study but rather a "necessary nuisance" on the way to tackling the actual research questions, we feel that a discussion of the parameterization challenges might draw the readers' attention too far away from the main topic. Nonetheless, we will provide further reasoning to indicate why a new parameterization and the new features were necessary.

Introduction: P2 second paragraph: "Nonetheless, it has proven difficult to identify clear relationships between forest structure and NPP (Chisholm et al., 2013) as several factors interact...": Suggest elaborating on the specific factors that complicate the relationship between forest structure and NPP.

Thanks, we will do that.

Method: Page 4 2.1 Field data: I have concerns about the allometry information and biomass equations provided in Supplementary A. (1) A lot of species lack allometry data and biomass equations. How did you address these species in your study? Did you use likelihood-based analysis similar to the PFTs classification? Please clarify this in the methods part. (2) The biomass equations, adopted from Chojnacky et al. (2014), are generalized primarily for North American species. Since most of the equations are empirical models, I doubt their accuracy when used directly.

We classified species into plant functional types (PFTs) to keep the number of free parameters tractable. Each PFT has a single set of "mean" allometric relationships, which we applied to all plants of the PFT – including those for which no specific allometry data were available.

In SI B.3, we explain how we fitted the allometric relationships to the available data. To mimic the Changbaishan forest as precisely as possible, we weighted each species' data according to the species' prevalence (basal area) in the inventory. Since the species for which data were available cover more than 96% of the basal area in the inventory, the missing data had a negligible effect on the resulting allometric relationships. Nonetheless, we will clarify that species with missing data were excluded in the process of fitting the allometric relationships.

The biomass equations referred to in SI A were only used to determine the aggregate biomass share of each PFT in the inventoried forest. This, in turn, was used to scale each PFT's mean stem biomass proportion to the correct value. This process is described in SI B.4.5. As we applied the biomass equations from Chojnacky et al. (2014) only on an aggregate level to estimate a single parameter per PFT, and since our approach is in line with existing literature (Piponiot et al., 2022), we believe that using the generalized equations does not undermine our results. In the actual model, we only used the allometric relationships that we derived directly from the field data.

Methods: Page 6 Model fitting, second paragraph, "We fitted these 18 parameters...": cannot get 18 parameters based on your description, please clarify.

Thanks, it should read "26 parameters" = 4 PFT-specific parameters * 6 PFTs + 2 general parameters. We will clarify this.

Results, first paragraph, "...This contrasts with the basal area of immature trees." please add supporting figures or statistical results.

The corresponding statistical results are provided in the next sentence. We will combine the sentences with a colon for better clarity.

Results: "...These results are depicted in Figures 4 and 5." Figures should be accompanied by the corresponding results in brackets. This will make it clearer for readers to correlate the interpretation with the figures.

Thanks, we will clarify what is visible in each individual Figure.

Results: "...These results are shown in Fig. 6." Similar suggestions as above.

We will number the figure's panels so as to be able to refer to them individually.

Results, Figure 7, does each dot correspond to a forest patch of 0.04ha? Please clarify.

Correct. We will clarify this.

Discussion, Page15, second paragraph, "The proportionality can be explained by the strong connection between the individual-level basal area and GPP in conjunction with the negligible NPP of mature trees." Consider simplifying or re-framing it for better readability.

We will do that.

Discussion, Page 15, second paragraph, "On the stand level, however, neither the GPP nor the respiration were correlated with the proportion of immature trees (Fig. 7) ...", in the discussion section, only this sentence refers to the figures. Please maintain consistency.

We will remove the reference to Fig. 7 to maintain consistency.

---

## Author Response (AR1)

**Responses to Reviewer 1**

**General Comments:**

The manuscript applied the existing individual-based forest gap model (FORMIND), which was developed using data from an old-growth temperate forest in the northeastern China. Authors present a novel methodology to distinguish between mature and immature trees to understand forest productivity. This approach offers a fresh perspective compared to traditional methods focusing mainly on gross primary production (GPP). The manuscript is well-organized, systematically presenting its research approach, results, and conclusions.

Manuscript is interesting and useful to international audience of the journal. However, there is room for improvement in the manuscript. The approach and conclusions are somewhat limited by the methodological framework and the absence of a comprehensive analysis of the ecological implications. Authors are suggested to address the following issues while making the revision.

> Thank you for your thorough review of our manuscript and the many helpful comments. Below, we respond to each of them and explain how we addressed them in our revision.
>
> Our responses are written in blue font and indented.

The introduction provides adequate background but lacks a critical review of previous research methodologies (other process-based models) and their limitations. Also, there is a need to introduce the limitation of the current FORMIND model. Suggest to enhance the introduction.

> Thank you for this suggestion. We have added a short review of other process-based models to the introduction (lines 78ff in tracked changes).
>
> Of course, FORMIND and other forest models have limitations, which are very important to understand to correctly interpret the results. In order to appropriately account for these limitations, it is important to be specific and explain how each individual limitation may affect the results. Otherwise, we would be at risk of leaving the reader with common sense statements about the limitations of modelling in general.
>
> We believe that the introduction, which we intend to set the frame of the study and to provide general guidance to the reader, is not the best place for a discussion of limitations with the necessary level of detail. Therefore, we provided a thorough discussion of limitations in section 4.1. We have extended this section further in response to your comments below.

Different species might exhibit significant variations in growth and carbon dynamics, even within the same maturity classification. The selection and classification of trees into mature and immature categories are not sufficiently justified. More rigorous criteria and a discussion of potential biases in these classifications are needed.

> Thank you for this thoughtful observation. We agree that a binary distinction of mature and immature trees could be difficult in field studies (as acknowledged in section 4.1; lines 473ff in tracked changes).

> Our maturity definition, based on the carbon use efficiency (CUE) that an individual could attain in the absence of competition, allows us to distinguish competition (which is production-related) and individual growth bounds (loss-related) as main limiting drivers of NPP. We admit that this may not have been sufficiently clear in the paper. We have therefore clarified and adjusted the respective sections in the introduction and discussion accordingly. Furthermore, we have added a discussion of the maturity definition and potential biases to the main part of the discussion section (lines 364ff in tracked changes).
>
> For a potential application in field studies, we suggested and analyzed the DBH entropy as a proxy for tree maturity. However, within the context of our modelling study, we could also identify the process (i.e., individual limitations rather than competition) that drives productivity.
>
> To account for the species- and individual-specific differences in carbon dynamics, we incorporated species-dependent and randomized growth limitations into the model.

The choice of the FORMIND model may not fully capture the complexity of forest dynamics, especially in terms of species-specific interactions and responses to environmental variables. For instance, you did not apply any space competition in the model. Suggest to compare results with those obtained from alternative models, particularly those incorporating more detailed species-specific parameters or interactions with abiotic factors.

> Thank you for this suggestion. The model *does* feature space competition, but we may have caused some confusion by misleadingly using the word "space competition" for "crowding mortality", which is typically modelled as increased (stochastic) tree mortality in "full" forest patches. We did not use this stochastic phenomenological model for space competition. Instead, as a new innovation, we took a more mechanistic approach and introduced tree mortality due to strong light competition. That is, trees that do not receive enough light to fulfill their respiratory needs die.
>
> This process has the same effect as crowding mortality ("full" forests lead to deadly overshadowing of small plants) but has a better mechanistic justification. In fact, both the species of the shadowing and the overshadowed trees are considered explicitly, since the LAI of larger plants as well as the respiratory demands of smaller plants are parameterized individually for each plant functional type (PFT). We have clarified this in the supplement section B.8 and changed the wording of "space competition".

The acknowledgment of the model's limitations is a positive aspect, but the discussion lacks a critical assessment of how these limitations might have influenced the study's conclusions. Suggestions for alternative modeling approaches or supplementary methods to address these limitations would provide a more balanced view.

> Thank you for pointing this out. We have extended the limitation section 4.1 to cover the effect of the limitations on the results more explicitly.

A lot of supplement information is provided with the manuscript. I'm not sure if it can refer to other fundamental literature previously published. Are the allometric relationships part of the FORMIND model? It would be better to keep concise and easier for the readers to understand.

> Thank you for this question and comment. Our study contains several innovations that were necessary to accurately reproduce the behaviour of the Changbaishan forest. The increased accuracy came at the cost of a longer paper supplement. The supplement also contains the

other values of the Changbaishan parameterization. This is required for full reproducibility of the study.

We understand that the length of the supplement poses a challenge to readers, who typically need to quickly identify the information most relevant to them. We have therefore introduced a table of contents and summary of our model extensions at the beginning of the supplement to guide the reader.

**Detailed comments:**

Introduction: Consider providing a brief introduction on any challenges or limitations encountered while adapting the FORMIND model to this specific old-growth temperate forest.

Thank you for this suggestion. As the parameterization is not the focus of our study but rather a "necessary nuisance" on the way to tackling the actual research questions, we feel that a discussion of the parameterization challenges might draw the readers' attention too far away from the main topic. Nonetheless, we have provided further reasoning to indicate why a new parameterization was necessary in section 2.2. Furthermore, we have added a high-level summary of the parameterization procedure to SI B, which also contains an overview and brief motivation of all major changes that we have introduced to the model.

Introduction: P2 second paragraph: "Nonetheless, it has proven difficult to identify clear relationships between forest structure and NPP (Chisholm et al., 2013) as several factors interact...": Suggest elaborating on the specific factors that complicate the relationship between forest structure and NPP.

Thanks, we have adjusted the section accordingly.

Method: Page 4 2.1 Field data: I have concerns about the allometry information and biomass equations provided in Supplementary A. (1) A lot of species lack allometry data and biomass equations. How did you address these species in your study? Did you use likelihood-based analysis similar to the PFTs classification? Please clarify this in the methods part. (2) The biomass equations, adopted from Chojnacky et al. (2014), are generalized primarily for North American species. Since most of the equations are empirical models, I doubt their accuracy when used directly.

We classified species into plant functional types (PFTs) to keep the number of free parameters tractable. Each PFT has a single set of "mean" allometric relationships, which we applied to all plants of the PFT – including those for which no specific allometry data were available.

In SI B.3, we explain how we fitted the allometric relationships to the available data. To mimic the Changbaishan forest as precisely as possible, we weighted each species' data according to the species' prevalence (basal area) in the inventory. Since the species for which data were available cover more than 96% of the basal area in the inventory, the missing data had a negligible effect on the resulting allometric relationships. Nonetheless, we have clarified that species with missing data were excluded in the process of fitting the allometric relationships and also made a respective note in the description of the data availability table.

The biomass equations referred to in SI A were only used to determine the aggregate biomass share of each PFT in the inventoried forest. This, in turn, was used to scale each PFT's mean stem biomass proportion to the correct value. This process is described in SI

B.4.5. As we applied the biomass equations from Chojnacky et al. (2014) only on an aggregate level to estimate a single parameter per PFT, and since our approach is in line with existing literature (Piponiot et al., 2022), we believe that using the generalized equations does not undermine our results. In the actual model, we only used the allometric relationships that we derived directly from the field data. We added a corresponding note to the data availability SI.

Methods: Page 6 Model fitting, second paragraph, "We fitted these 18 parameters...": cannot get 18 parameters based on your description, please clarify.

Thanks, it should read "26 parameters" = 4 PFT-specific parameters * 6 PFTs + 2 general parameters. We have corrected this.

Results, first paragraph, "...This contrasts with the basal area of immature trees." please add supporting figures or statistical results.

The corresponding statistical results are provided in the next sentence. We have combined the sentences with a colon for better clarity.

Results: "...These results are depicted in Figures 4 and 5." Figures should be accompanied by the corresponding results in brackets. This will make it clearer for readers to correlate the interpretation with the figures.

Thanks, we have referenced each individual panel for each result in the revised version.

Results: "...These results are shown in Fig. 6." Similar suggestions as above.

Again, we have referenced each individual panel for each result in the revised manuscript.

Results, Figure 7, does each dot correspond to a forest patch of 0.04ha? Please clarify.

Correct. We have clarified this.

Discussion, Page15, second paragraph, "The proportionality can be explained by the strong connection between the individual-level basal area and GPP in conjunction with the negligible NPP of mature trees." Consider simplifying or re-framing it for better readability.

We have rephrased the sentence as follows: "The proportionality can be explained by noting that (1) the individual-level GPP is strongly positively correlated with basal area, irrespective of the maturity stage, and (2) only immature trees contribute to the NPP."

Discussion, Page 15, second paragraph, "On the stand level, however, neither the GPP nor the respiration were correlated with the proportion of immature trees (Fig. 7) ...", in the discussion section, only this sentence refers to the figures. Please maintain consistency.

We have removed the reference to Fig. 7 to maintain consistency.

**Responses to Reviewer 2**

**General Comments:**

Fischer et al. present a novel method for relating the net primary production (NPP) of forests to forest structure. They explore this method using the process based forest model FORMIND parameterised with data from a forest inventory plot located in an old growth temperate forest in Changbaishan, China. The method rests on the assumption that as trees grow their respiratory demands increase, resulting in a decline in NPP and carbon use efficiency with size. As a result, the proportion of immature trees in the forest predicts stand level NPP as only immature trees are putting on biomass. The authors test a number of structural metrics and find that a measure of DBH entropy is the best proxy for the proportion of immature trees and could thus be used to predict the NPP of a forest using only inventory data.

This topic will be of interest to the forest ecology community and those interested in predicting carbon sequestration using metrics of forest structure. The manuscript has a good structure and is for the most part easy to follow. The figures show the results clearly. I think keeping most of the details of FORMIND in the supplement and only describing necessary details in the main manuscript is a good approach, although I think some more high level summaries in the main text would be helpful so that readers do not have to keep moving to the supplement which is very long.

> Thank you for your thoughtful review and helpful comments! Below we respond to each of your points and explain how we have addressed them in the revised manuscript.
>
> We understand and agree that the supplement is long, potentially making it difficult for readers to find the information they are interested in. To mitigate this issue, we have added a table of contents and a high-level summary to the supplement B. Furthermore, we have added information on the main mechanisms of Formind to section 2.2.

My main suggestion is to include more discussion of how some of the assumptions in FORMIND influence results. In this manuscript, FORMIND is set up so that trees have a maximum DBH, above which they allocate all GPP to respiration meaning that NPP goes to zero. It is therefore not surprising that the proportion of immature trees predicts NPP. The authors acknowledge in the discussion that there is still debate in the literature about whether NPP declines with size. It would be good to expand on this and discuss how uncertainty in the changes of individual NPP with size influence the ability of stand structure to predict stand level NPP. They could also test this assumption using the field data by looking at a time series of growth rates for individual trees and identifying declines in growth with tree size.

> Thank you; this are excellent points.
>
> Letting the NPP go to zero with increasing tree size is a long-standing modelling paradigm of forest models and not easy to change ad hoc. Nonetheless, following your comment, we have thought of a way to assess the effect of the zero-NPP assumption and to test milder versions of it. We ran the model assuming that mature trees (though not growing) still have a positive NPP, given as a certain fraction of the NPP of immature trees of the same PFT and size. We varied this fraction and found that already reducing the CUE by only 25% allowed us to observe the effects we presented in the paper. We included these simulation experiments to both the main text (methods, main results, discussion) and the supplement (detailed results).

Unfortunately, it is difficult to test the low-NPP assumption directly on our field data, as tree maturity likely depends on local conditions we do not have data for. Furthermore, the inventory data are partially subject to measurement errors (manifesting in negative or very large DBH increments) making us a little sceptical regarding such a detailed (individual-focused) direct analysis of the field data. Nonetheless, we present DBH-dependent DBH increment data on an aggregate level in SI B.7.3.

**Specific comments:**

It would have been helpful if the manuscript had line numbers.

We totally understand that and are sorry that we could not deliver a manuscript with line numbers due to regulations of the preprint server we used (ArXiv). We have now provided a version with line numbers.

On page two the authors say that CUE is expected to decline with size because large trees have higher demand for respiration and non-structural carbon. This doesn't necessarily follow unless GPP increases with size are less than respiratory and NSC increases. Or GPP asymptotes or decreases. Do we know this to be the case in this forest?

We do not have data on the CUE of individual trees in this forest. Therefore, we modelled the CUE of trees based on the assumptions laid out in lines 154ff (tracked changes).

Our statement was primarily motivated by theoretical arguments. Increases in GPP require new structures such as leaves, which in turn need to be maintained. The higher trees grow the smaller is the additional gain of additional leaves due to self-shading and the higher are the additional costs due to longer transport pathways. We are not aware of a study suggesting an increasing CUE for individual trees. This is a relevant point, and we would love to consider this here or in follow-up studies if corresponding data are available.

Bottom of page 4 - missing word - "*The* key idea".

The corresponding sentence has been removed in response to another comment.

Page 5 - how were the light requirements of the species known? Do those classifications align with growth and survival rates from the inventory data? It looks like in Fig S1 the two species with the fastest DBH increment rates are on almost opposite sites of the shade tolerance spectrum which is a little surprising. Was there any comparison of growth rates predicted by FORMIND and the mean growth rate per PFT from the inventory data?

Thank you for these observations and questions!

The light requirements were derived from available literature such as *Niinemets & Valladares (2006) and Wang et al. (2010). The provided values correspond roughly to the minimal required sunlight for growth of saplings: a value of* 1 corresponds to a light requirement of >50% of the full sunlight; 2: 25–50%; 3: 10–25%; 4: 5–10%; 5: 2–5%. We have clarified this in the revised manuscript.

Though the inventory data are the best we have, the sample sizes for rare species are small. This applies in particular to the species with the highest DBH increments. We agree that the combination of high median DBH increment and high shade tolerance is surprising for Abies nephrolepis, and it was classified as a mid-tolerant species. A different classification of the

> species would be possible but hardly affect the parameterization and results, as Abies nephrolepis is rare in the Changbaishan area (see SI A).

> Unfortunately, the sample size issue impact also the mortality analysis. Nonetheless, we have analyzed the mortality rates for each PFT and presented the results in SI B.9 (see also Fig. S11). There it is visible that the mortality of shade intolerant species is mostly higher than the mortality of mid-tolerant and shade-tolerant species.

Please provide the Genus name of Q. mongolica in the first mention of this species.

> Done.

I didn't understand how growth was modeled without reading the SI. Consider some high level descriptions in the main text.

> Thanks; we have added a description of the main processes governing tree growth in Formind to the beginning of section 2.2.

Middle of page 6 - why did the authors use biomass and stem count as an indicator of the size distribution rather than the actual size-distribution?

> As our goal was to correctly model the forest dynamics, driven by local competition, we desired to match the local forest composition. At the local scale (here: 20m x 20m), the forest composition can show different patterns than on the larger scale. For example, a forest may consist of 50% species A and 50% species B, but aggregation and segregation may lead to patches with, say, 75% species A / 25% species B and vice versa, meaning that the species are not well-mixed. This local structure is important for understanding the forest dynamics but difficult to cover with aggregate measures.

> Since a forest's local composition can be variable, we needed to apply a stochastic framework to fit the model (which is in line with FORMIND's stochastic nature anyway). To that end, we needed to estimate the *distribution* of the local forest states. That is, for example, how likely will we find a local patch with 75% species A and 25% species B? How often will we find 50% / 50%? The joint distribution of trees and their sizes has a high dimension, making it challenging to estimate this distribution. Considering, say, 10 different size classes for each PFT would lead to a 60-dimensional estimation problem, which is practicably infeasible (unless someone comes up with exciting new sampling / estimation theory). Therefore, we considered summary statistics of the distributions of biomass and stem count at the scale of 20m x 20m. We have added further clarification to the text (lines 195ff in tracked changes).

> Preliminary simulation experiments with additional statistics (not shown in the paper due to space limitations; this would be a full second paper) indicated that the information loss due to summary statistics is small, and the gain via improved likelihood estimates (due to a smaller-dimensional sampling space) is larger, which helps us finding the true likelihood maximum.

There are not many details of wood decomposition or soil respiration parameterisation in the methods section. The authors could consider adding a few sentences explaining this aspect of FORMIND. It wasn't really clear to me what the hypotheses were for how NEE would change with forest structure.

> Thanks; we have added information on this process to the manuscript.

Page 9 section 2.5. To identify trees that have reached growth limits in field data would it not be simpler to look at growth rates and find large trees that have declining growth rates (if there are enough census intervals, or if not, those that are growing less than some quantile of the population)? Or is the idea that DBH entropy can be used in field studies with a single census?

> Thank you for these ideas! In our dataset it was difficult to clearly identify declining growth rates, also due to the limited number of census intervals. If we considered all trees that are growing less than similar conspecifics, we would still need to distinguish the causes for the reduced growth: is it due to production-related factors (e.g. reduced light / shading) or due to internal limitations (maturity)? We have added this point in the discussion section.
>
> It is correct that we suggest using DBH entropy as a proxy in field studies, and yes, it can be used with a single census.

Fig 4. Why are there few forest patches with a Shannon diversity of just over 1?

> Interesting question! We would like to start by observing that the Shannon diversity is computed based on discrete numbers (stem counts and PFTs). Therefore, the Shannon diversity is discrete itself and not continuous, and there is no a priory reason to expect that neighbouring diversity values should occur with similar frequency.
>
> Now we consider the specific question why Shannon diversity values just above 1 occur much less frequently than values of exactly or a little less than one. A Shannon diversity of 1 is obtained if a patch contains exactly two PFTs with the same stem count. If the stem counts have a small relative error (which, noteworthily, requires that there are many stems), the Shannon diversity is just a little smaller than one.
>
> A Shannon diversity just over 1 requires that there are three species: two with almost equal, large stem count and one with small stem count. In our simulations, such species compositions occurred rarely and, in particular, much less frequently than species compositions with two almost equally frequent species. Ecologically, this may be a sign of conspecific aggregation.

Results in Fig. 4 are not that surprising since the model was set up so that NPP would not be affected by large trees because large trees allocate all GPP to respiration. But there is no test of that in the field in this study. Is that a reasonable assumption? Some evidence suggests that large trees continue to actively accumulate biomass e.g. Stephenson et al. 2014 https://www.nature.com/articles/nature12914. Ah, I see the authors bring this up later in the discussion. If possible the authors should try to address that assumption using the inventory data from Changbaishan.

> Thank you for bringing up this point! We have addressed it in our response to your main suggestion above (second response) and found that our results still hold if our allocation assumption is significantly weakened. An ad-hoc analysis of the field data did not yield conclusive results, partially due to missing data on the growth potential of the individuals at their specific sites and uncertainties in individual allometries and repeated DBH measurements. A more thorough analysis could potentially address some of these points but was beyond of the scope of this study.

Fig 4. NEE is very low in mature patches - is this because of less wood turnover? Is all CWD in FORMIND the result of mortality or is there also a representation of branch turnover?

Thank you for this observation. A small (or very negative) NEE corresponds to large carbon emissions from the forest (which may result from *increased* wood turnover). Forests with many mature trees have a small NPP, which also manifests in a low NEE.

In FORMIND, NEE is dominated by mortality but also includes mortality of branches falling dead in response to a negative NPP (i.e., if the respiratory needs cannot be satisfied).

Bottom of page 11. Does respiration here include soil respiration? Same comment for Fig. 7c.

No, here (and in Fig. 7c) respiration refers to the respiration and other carbon losses of alive trees only. We have clarified this in the revised paper.

**References:**

Niinemets, Ü. and Valladares, F. 2006. Tolerance to shade, drought, and waterlogging of temperate Northern Hemisphere trees and shrubs. Ecological Monographs 76:521–5471.

Wang X, et al. 2010. Spatial distributions of species in an old-growth temperate forest, north-eastern China. Canadian Journal of Forest Research, 2010, 40: 1011-1019.

---

## Author Response (AR2)

**General Comments:**

The authors have addressed my previous comments. In particular the additional analysis testing how sensitive results are to the loss of CUE in mature trees was helpful for demonstrating the robustness of the results.

> Thank you for reviewing our manuscript again and for your positive feedback. We are happy that the changes we implemented based on your helpful comments were suited to demonstrate the robustness of our results.

I have only a few minor comments at this stage:
1. Several times the authors mention that results will lead to new hypotheses or theory e.g. L 536 and L23. It would be useful to provide a few examples.

> Thank you for pointing this out. These hypotheses refer to the results we presented in the paper (finally summarized in the "Conclusions" section), which could inspire further field studies, as pointed out in section 4.2 ("Outlook"). In the revised manuscript, we explicitly mention two main hypotheses for better clarity (see the end of the paper).

2. L339 - should that be mature trees?

> Yes, this is correct. We have fixed this. Thank you!

**Additional Note:**

> We noticed that our previous submissions were subject to an embarrassing glitch: we forgot to uncomment an abstract draft that we incautiously wrote directly under the "Conclusions" section. We have now deleted the additional text, which was only reiterating the contents of the paper.

---

## Author Response (AR3)

Dear Editorial Team,

Thank you for carefully checking our manuscript and providing helpful comments improving the consistency and accessibility of our paper.

In response to your comments, we have

1. Relabelled the items in the supplement's table of contents, which included the misleading prefix "Appendix" in each line.
2. We changed the colour scheme of all line plots with more than two line colours to a scheme more accessible to colorblind readers. We furthermore added legends in Figs. S1 and S2 so that colorblind readers do not need to refer to the verbal description of the colors. In these figures, we have also introduced different marker shapes to help colorblind readers distinguishing the data groups.
3. We revised the numbering of the figures in the supplement to a consistent and consecutive scheme.
4. We fixed a LaTeX error hindering the tables in the supplement from being displayed.

The errors we fixed here (points 1, 3, and 4) were all introduced when we attempted to apply the journal's LaTeX class (in the second revision, whose content is almost identical to the first revision). Hence, all information potentially changed here were present and reviewed before.

If there are any further items that can help improving our manuscript, please do not hesitate to let us know.

Best wishes,

Samuel Fischer
on behalf of the authors